# Neurological and Inflammatory Manifestations in Sjögren’s Syndrome: The Role of the Kynurenine Metabolic Pathway

**DOI:** 10.3390/ijms19123953

**Published:** 2018-12-08

**Authors:** Fabíola Reis de Oliveira, Marina Zilio Fantucci, Leidiane Adriano, Valéria Valim, Thiago Mattar Cunha, Paulo Louzada-Junior, Eduardo Melani Rocha

**Affiliations:** 1Ribeirao Preto Medical School, Ribeirao Preto, University of Sao Paulo, Ribeirao Preto, SP 14049-900 Brazil; fabiolabi@hotmail.com (F.R.d.O.); marinazf@fmrp.usp.br (M.Z.F.); leidianeadriano@hotmail.com (L.A.); thicunha@fmrp.usp.br (T.M.C.); plouzada@fmrp.usp.br (P.L.-J.); 2Espírito Santo Federal University, Vitoria, ES 29075-910, Brazil; val.valim@gmail.com

**Keywords:** IDO, kynurenine, pain, Sjögren’s syndrome, tryptophan

## Abstract

For decades, neurological, psychological, and cognitive alterations, as well as other glandular manifestations (EGM), have been described and are being considered to be part of Sjögren’s syndrome (SS). Dry eye and dry mouth are major findings in SS. The lacrimal glands (LG), ocular surface (OS), and salivary glands (SG) are linked to the central nervous system (CNS) at the brainstem and hippocampus. Once compromised, these CNS sites may be responsible for autonomic and functional disturbances that are related to major and EGM in SS. Recent studies have confirmed that the kynurenine metabolic pathway (KP) can be stimulated by interferon-γ (IFN-γ) and other cytokines, activating indoleamine 2,3-dioxygenase (IDO) in SS. This pathway interferes with serotonergic and glutamatergic neurotransmission, mostly in the hippocampus and other structures of the CNS. Therefore, it is plausible that KP induces neurological manifestations and contributes to the discrepancy between symptoms and signs, including manifestations of hyperalgesia and depression in SS patients with weaker signs of sicca, for example. Observations from clinical studies in acquired immune deficiency syndrome (AIDS), graft-versus-host disease, and lupus, as well as from experimental studies, support this hypothesis. However, the obtained results for SS are controversial, as discussed in this study. Therapeutic strategies have been reexamined and new options designed and tested to regulate the KP. In the future, the confirmation and application of this concept may help to elucidate the mosaic of SS manifestations.

## 1. Introduction

Sjogrën’s syndrome (SS) is defined as an exocrinopathy of the salivary and lacrimal glands (SG and LG) mediated by autoimmune mechanisms that could manifest neurological dysfunctions, and those neurological dysfunctions may take part in the physiopathology of the disease [1,2,3,4,5]. However, the extraglandular manifestations (EGM) of neurological disorders are not considered in the definition or the diagnosis of SS, despite their presence during the disease progress evaluation and reported more frequent association with SS in recent years [6,7,8,9]. Of interest, 60–80% of patients develop neurological manifestations before or at SS diagnosis (early systemic presentation), indicating that neurological damage is precocious and it could play a role in the disease mechanism [10] (Figure 1).

The kynurenine metabolic pathway (KP) is the main pathway that is involved in the catabolism of tryptophan. There is evidence that KP participates in the inflammatory mechanisms of the neurogenic manifestations of autoimmune diseases through the action of indoleamine 2,3-dioxygenase (IDO), the rate-limiting enzyme in tryptophan degradation [11,12,13,14,15].

This review summarizes the actual status of knowledge concerning the neurological manifestations in SS and it presents the hypothesis of the association between these neurological changes and the KP and their interactions to understand the unknown and paradoxical signs and symptoms of SS (Box 1). 

Box 1.Summary of evidence linking Sjögren’s syndrome (SS) and the tryptophan/kynurenine signaling pathway (KP) in the central nervous system (CNS).
Association among chronic inflammation, pain and neuropathic disorders in SS [16]Dryness and Indoleamine 2,3-dioxygenase (IDO) activity triggered by interferon [17,18]Clinical findings and inflammation modulated by sex hormones in SS [19]Tryptophan deprivation induces dry eye [20]Sjögren’s syndrome and salivary gland inflammation leads to increased expression of kynurenine, a metabolite of IDO [18,21]


## 2. Autoimmunity, Neuropathy and Chronic Pain

The constant basal and reflex wetting of the mouth and the ocular surface provided, respectively, by saliva and by tears are directly controlled by the autonomic nervous system [22,23,24]. The volume and content of fluids from several exocrine glands that are present in both locations (mouth and eye) are responsive to sensorial stimuli from the environment that is driven by sensitive nerves specialized in taste and vision but also general sense nerves related to touch, thermal and chemical changes [25,26,27]. Once detected by the brainstem, the feedback mechanisms are conducted in the parasympathetic and sympathetic systems, stimulating synapsis in α-adrenergic and muscarinic cholinergic receptors in the many salivary and lacrimal gland subtypes distributed in the mouth and in the ocular surface [25,28]. This sensorial/autonomic feedback system that regulates the tear secretion in the ocular surface is called the lacrimal functional unit (LFU) (and a very similar system works in the mouth), whereas inflammation, trauma, or other damage in a segment of this integrative system can disrupt persistent dysfunction of secretion with variable sensorial manifestations, depending on the integrity of the sensorial loop [28,29].

Autoimmunity is linked to chronic neuropathy in at least three different domains in SS. First, the major clinical EGM observed in SS are fatigue and pain, with different manifestations, such as allodynia, dysesthesia, hypo or hyperestesia, and hyperalgesia [9,30,31]. Second, chronic inflammation in the target organs (e.g., the ocular surface) generates a noxious stimulus and depression that may persist in a further phase in which the inflammatory process is already resolved [32,33]. Third, the central nervous system (CNS), mostly autonomic nervous system dysfunction, can induce or perpetuate an unbalanced inflammatory response in the target-innervated organs of autoimmune diseases [3,34]. Therefore, an unified theory of the relationship between the neural and immune mechanisms of SS manifestations take in account that lesions of the autonomic and peripheral neural system reduce the threshold for inflammatory and noxious events in their connected organs and disturb the balance between pro and anti-inflammatory mediators [35,36,37] (Figure 2). 

The chronic pain and persistent inflammation that are mediated by humoral factors and neurotransmitters associate autoimmunity and neuropathy in several target organs, including the hippocampus and lacrimal functional unit (LFU) in SS [27,34,38,39,40,41]. The dissociation between signs and symptoms in dry eye and dry mouth disease, including SS, delays and makes the diagnosis more difficult [5,42,43,44,45,46,47,48]. The lower intensity of signs that are associated with depression and diffusely spread painful symptoms in the body suggest a mechanism involving integrative systems, such as the autonomic nervous system, immune, and endocrine systems (Figure 1) [42,45,49]. 

The pain, depression, and other neurological manifestations of SS can be associated with IFN-γ inducible KP activity. Therefore, our hypothesis considers three aspects: (a) the KP in the mechanism of SS neuropathy; (b) the possible roles of the KP in cross-talking pathways among the immune, endocrine, and neural systems to produce the SS signs and symptoms; and, (c) the possible role of the KP in the previously reported dissociation between those signs and symptoms in SS [50,51,52,53,54,55]. Therefore, IFN-γ-inducible KP could be the missing link between disease activity and neural manifestations in SS [56] (Figure 2). 

In summary, this hypothesis implicates the KP in the spectrum of manifestations of SS, with two different poles of SS disease. At one pole, the major characteristic is chronic noninflammatory pain with neuropathic features. At the opposite pole is SS present with inflammation, including EGM inflammatory activity. In other words, we hypothesized that SS patients with Interferon-γ (IFN-γ)-inducible KP activation could develop chronic pain, depression, and low EGM disease activity, because the KP promotes an immunosuppressive and neurosensitive effect. However, these distinguished profiles of SS are more clearly documented in long-term cohort studies, as recently published [16].

### 2.1. SS and Neurological Manifestations 

Neurological signs and symptoms have been described in SS. The prevalence of peripheral neuropathy ranges from 8–49%, depending on the selection bias related to different classification criteria, and whether neurological manifestations were diagnosed based on clinical marked symptoms versus asymptomatic, or detected by electrophysiological studies [2,57,58,59,60,61]. Distal sensory and sensorimotor neuropathies are the most common manifestations of peripheral nerve disease in primary SS (pSS). Sensory neuropathies include painful nonataxic sensory polyneuropathy, small fiber neuropathy, dorsal root ganglionitis, and trigeminal neuropathy. Other forms have also been described, including multiplex mononeuritis, acute or chronic inflammatory demyelinating polyradiculoneuropathy (CIDP), multiple cranial neuropathy, especially III, V, VI, VII, IX, X, and XII, and dysautonomia, which is very common in SS and can reach a frequency of 40% of SS patients, with or without different neurological manifestations [62,63,64,65]. Moreover, CNS manifestations of SS may be associated with focal syndromes (such as multiple-sclerosis-like epilepsy, movement disorders, neuromyelitis optica, and pseudotumor) and diffuse syndromes (encephalitis, meningitis, cognitive dysfunction, psychiatric disorders). Acute or chronic myelopathy and inferior motor neuron disease may also occur [57,58,62,66,67,68,69,70,71,72,73,74,75,76,77,78,79,80,81,82,83,84,85,86,87,88,89]. This high variability of EGM and responses to treatment in SS are not completely understood. The spectrum of the disease may be caused by environmental agents that lead to epigenetic phenomena working on diverse and complex mechanisms of organ injuries, such as vasculitis and lymphocytic infiltration [90,91,92]. In addition, the changes in SS patients in the CNS and PNS (such as in the dorsal spinal cord and the dorsal ganglion roots), including autonomic dysfunction, diffuse decreases in white matter, and loss of gray matter in the hippocampal area, may influence peripheral organ function and induce neurological symptoms [93,94,95].

### 2.2. SS and the Mechanisms of Neurological Manifestations

The mechanisms triggering the neurological manifestations in SS are unclear. They involve genetic predisposition, environmental agents, trauma and posttraumatic stress, autoimmunity against the CNS, and peripheral nervous systems (PNS), in addition to neuroimmunendocrine network disruption [60,90,96,97,98,99,100]. The events are attributed to DNA demethylation, microRNA abnormal expression, an imbalance of interferon I (α and β) and II (γ), and anti-neuron autoantibody production [68,90,101,102,103]. In SS, the 2-5 oligo-adenylate synthetase 1 (OAS1) gene defect leads to a reduced responsiveness to IFN-γ and higher production of IFN-γ, causing severe complications, such as lymphoma, neuropathy, and debilitating fatigue [100]. 

The sensorial, autonomic, cognitive neurological, or behavioral manifestations of SS are associated with structural changes, such as increases in corneal nerve thickness and a larger number of antigen-presenting cells in the cornea, nerve vasculitis, nonvascular encephalitis, neuromyelitis, the CNS, and axonal degeneration [38,60,97,104].

It is interesting to note that structures that are altered in the CNS, PNS, exocrine glands, and cornea (based on image exams) of SS patients are also sensitive to changes in the KP caused by trauma or ischemic damage (Table 1). These observations suggest that common inflammatory events are shared in autoimmune and non-immune inflammatory diseases, such as the synapses of the sensorial fibers, in the dorsal ganglion root, hippocampus, thalamus, and LFU [2,12,13,20,29,40,53,93,105,106,107,108] (Table 1). Those locations must be investigated in future studies addressing the anatomic correlations of clinical and biochemical changes in SS. 

### 2.3. Immune and Endocrine Modulation of Neurological Findings in SS

The concept that is implicit in the neuroimmunendocrine network predicts that cellular and molecular communication among those three systems are responsible for the homeostasis of the body organs, and a disruption in this network plays a role in the disease mechanism [112,113,114,115]. For example, acetyl-choline (Ach), dopamine, glutamatergic, and other neurotransmitters are secreted not only by neurons, but also by lymphocytes [116]. In contrast, the autonomic nervous system is capable of modulating lymphocyte proliferation in target organs, such as spleen, liver, kidney, and brain [37,117,118]. In addition, the sensory neurons are able to secrete peptides with immunomodulatory properties, such as galanin, netrin-1, and somatostatin and promote or attenuate inflammatory responses [119,120]. 

Hormones, in particular, sex hormones, can modulate inflammation and pain, sensitizing the ionic receptors expressed in neurons and epithelial cells, called the transient receptor of potential (TRP), and stimulating growth factor and cytokine expression in target tissues, such as the lacrimal and salivary gland (LG and SG), hippocampus, and trigeminal ganglion of the CNS, as well as other target tissues [19,121,122,123,124,125,126,127]. These mechanisms explain the sex hormone-mediated amplification and perpetuation of the inflammatory process, pain hypersensitivity, and exocrine gland dryness manifestations in SS, where estrogen potentiates the pain and pro-inflammatory mediators and androgens work in an opposite manner [19,96,128,129,130]. Although the trigger of the first event in SS and the steps leading to the chronic phase are poorly understood, there is a strong clinical association between female sex hormones and the inflammatory mediators that are involved in innate as well as adaptive immunity [4,19,57,124,131,132] (Figure 1).

Therefore, hormones and neural pathways interact with immune responses, acting on pain sensation, the inflammatory reaction, tissue integrity, and organ functional disruption [5,37]. These interactions certainly take part in the EGM and comorbidities in SS patients [2,106,133].

A loss or damage of nerve fibers, dorsal root ganglionitis, nerve vasculitis, and reduction of the CNS matter can be observed in imaging exams, such as magnetic resonance images (MRIs) or skin biopsies of SS patients; they are implicated in the underlying mechanisms and they could potentially function as diagnostic markers of neurological manifestations of SS [38,93,94,109]. For example, cognitive impairment in SS patients is associated with antibodies against the subtype 2 of the N-methyl aspartate receptor (NR2) (anti-NR2 antibodies) in the cerebrospinal fluid (CSF) mediating hippocampal gray matter atrophy, as observed by MRI [40]. 

In summary, the SS inflammatory activity in the PNS and CNS causes the above-described signs and symptoms, and the physiopathology implicates the hormones, neurotransmitters, and cytokines that are susceptible to the KP interference, as previously described [13,14,114]. The role of kynurenine and its metabolites in the neurological activities are detailed below. 

## 3. Kynurenine Pathway (KP)

The metabolism of L-tryptophan (LTF) leads to the generation of several neuroactive compounds via the serotoninergic and kynurenine pathways. In the serotoninergic pathway, L-tryptophan is metabolized to serotonin (5-hydroxitryptamine or 5-HT), and, in some cells, to melatonin. Serotonin acts as a neurotransmitter with a variety of functions in behavior and psychiatric symptoms (depression and anxiety), platelet aggregation, gastrointestinal tract control (satiety, secretion, and peristalsis), and tumor resistance. Melatonin acts as a neurohormone in the circadian rhythm, inducing sleep, and it also has anti-inflammatory, anti-angiogenic, and anti-tumoral immunomodulatory effects [134] (Figure 3). 

In the catabolic KP, tryptophan 2,3-dioxygenase (TDO) metabolizes tryptophan in the liver and responds to hormonal inputs, such as cortisol and glucagon, and to the tryptophan load in meals [135] (Figure 3).

The IDO enzyme regulates both innate and adaptive immune responses through degradation of the essential amino acid tryptophan into kynurenine and other metabolites, which suppress the effector T-cell function and promote the differentiation of regulatory T cells [136]. IDO also metabolizes serotonin, melatonin previously generated in the serotonergic pathway, and tryptophan to N-formyl kynurenine, and kynurenine, mostly in the lung, brain, and blood, producing quinolinic acid and nicotinamide adenine nucleotide (NAD+) [136,137,138]. The KP is responsible for 95–99% of tryptophan catabolism [139] (Figure 3).

IDO is found mainly in immune cells, and it has enzymatic activity in the cytoplasm and transcriptional activity in the nucleus, playing a unique role as a signaling molecule modulating immune responses [143,144,145]. The nuclear effect contributes to enzymatic “self-amplification” in an IFN-dependent loop that may account for the tolerance phenotype, attenuating or preventing immune reactions and mediating persistent pain in several conditions, including SS [18,143,144,146,147]. IDO activity is induced in macrophages by cytokines, such as IFN-γ and TNF-α and by prostaglandins. During viral and bacterial infections, lipopolysaccharides can trigger IDO activity in dendritic cells and enhance the subproducts of kynurenine [148,149,150,151,152]. IDO overexpression has been documented in patients with systemic lupus erythematosus (SLE) and SS, as well as in sepsis [15,50,153]. In patients who are positive for the IFN gene expression signature, Treg cell levels are elevated in combination with increased IDO activity, with tolerance and immune modulation [18]. These regulatory T lymphocytes represent a diverse subclass of T cells, a protagonist in the maintenance of self-tolerance and immune modulation [154].

The KP is catalyzed by the IDO enzyme and it can be induced by cytokine IFN-γ, as observed in studies in mouse, in which IFN-γ or IFN-γ receptor knockout prevents kynurenine production [155]. This phenomenon is dependent on the abovementioned antigen-presenting cells (APCs), as observed in mouse models of graft-versus host disease (GVHD), and the induction of this KP can extend the life-span and reduce inflammation in the gut of wild-type mice, but not IFN-γ receptor knockout, mice [155]. 

The balance between Interferon-alpha and beta at one side and gamma at the other (INF-α and β, and IFN-γ) activation in inflammatory processes can modulate the activation of KP and the resulting intensity of this inflammatory process, not just in SS but also in response to other stimuli, such as exogenous challenges [156,157,158]. IFN-α and β (type I INF) are associated with inflammatory activity (including higher levels of autoantibodies and inflammatory cell activity), and IFN-γ (type II INF) is associated with KP activation and the attenuation of inflammatory activity [91,159,160,161]. SS patients have higher levels of IFN-γ and INF-α and β mRNA in the peripheral blood and labial salivary glands in comparison to healthy individuals, and SS patients with lymphoma present lower levels of IFN-α and higher levels of INF-γ mRNA in labial salivary gland samples when compared with other SS and control individuals, supporting INF-α/γ as a predictor of lymphoma in SS [156]. Therefore, type I IFN seems to have an antagonistic effect to type II IFN, but an unclear, though potentially useful, combinatorial effect in SS associated with lymphoma. 

The possibility that some cases of SS are disrupted by viral infection also raises the possibility of a subtype of SS that is triggered by viral disease and distinguished KP activity, with a particular set of signs and symptoms. In support of this possibility, 50% of pSS patients have been confirmed to have hepatitis delta virus, without hepatitis B virus, in the salivary gland in a recent report, and the affected patients presented elevated SG inflammation and autoantibody positivity [162].

The isoform IDO2 as well as TDO1 are enzymes that are involved in the catabolism of the amino acid tryptophan and operate in similar manners in immunomodulation, with variations in the target tissues and cells involved in the metabolic pathways. However, further studies are necessary to clarify their diversity in diseases, including SS [163,164].

## 4. KP and Neurological Manifestations

The relationship between tryptophan, serotonin, and depression has an extended history in psychiatry. The development of depressive symptoms is correlated with high levels of tryptophan metabolites in urine [51] and with a decrease in tryptophan in the blood and cerebrospinal fluid [165,166]. Tryptophan transport across the blood-brain barrier, specific inflammation, and damage that are caused by brain-reactive autoantibodies and immune complexes play a critical role in the regulation of tryptophan metabolism in the brain [134]. There is substantial evidence to suggest that, in addition to serotonergic neurons, other cells, such as astrocytes, dendritic cells, microglia, and macrophages also synthesize multiple neuroactive metabolites via the enzyme IDO and KP in the CNS [167]. Evidence for the underlying mechanisms have been obtained from clinical studies examining the effects of IFN-α on the mood of cancer and hepatitis C virus-infected patients. In both cases, the development of depressive symptoms was associated with decreased circulating tryptophan levels and the enhanced formation of kynurenine, indicating activity of the IDO pathway [168,169,170].

Tryptophan metabolites that are generated in the KP have been associated with neurodegenerative diseases, such as acquired immunodeficiency syndrome (AIDS)-related dementia, Alzheimer’s and Huntington’s diseases, and neuropsychiatric diseases, such as bipolar disorder and schizophrenia [171,172]. The synthesis of kynurenine in the CNS is affected by dietary intake of tryptophan and by the gut microbiota [14,173]. Tryptophan deprivation induces depression and cognitive dysfunction (attention, memory, and execution), among other neurological dysfunctions, through the glutamatergic receptors [14,54,173,174,175,176].

Tryptophan metabolism induces a dual effect in astrocytes and microglial cells [177]. In astrocytes, it leads to the production of kynurenic acid, which has been reported to participate in neuroprotective effects. In contrast, microglial cells give rise to metabolites with reactive oxidative properties, including hydroxykynurenine and 3-hydroxyanthranilic acid and quinolinic acid, which also functions as an agonist of the glutamate N-methyl-D-aspartate (NMDA) receptor subtype and may contribute to excitotoxicity and neurotoxicity [177,178,179].

In healthy subjects, a well-adjusted system is established in the KP via the action of kynurenines aminotransferases (KATs) on kynurenic acid or quinolinic acid by (KMO) [141]. Kynurenic acid has been reported to promote neuroprotective and immunosuppressive actions in the CNS and plays a role as an NMDA antagonist, blocking this glutamatergic receptor. Enhanced IDO activity and a deviation to KMO downstream are observed by stimulation of cytokines, such as IFN-γ [141]. Additionally, SS1 patients with positive levels of IFN-γ in the blood have higher levels of CD25 FoxP3+ Treg cells, which correlate with higher plasma IDO activity, as measured by the tryptophan/kynurenine levels, in comparison to the controls [18]. KMO has a clear inflammatory and pro-apoptotic action. Quinolinic acid, its intermediate compound, acts as an agonist at the NMDA receptor, modulating excitatory amino acid transmission, and it may serve as a neurotoxic agent implicated in the pathogenesis of several neurological diseases [12,134,171,179]. Moreover, in SLE, the presence of the antibody against the NMDA receptor has several demonstrated pathological activities in the kidney and brain, particularly in the hippocampus, including neuron death, as observed in humans and in mouse models [180,181]. Likewise, 3-hydroxykynurenine also has a neurotoxic effect, which is probably associated with the conversion of reactive oxygen species and apoptosis [94,140,182,183,184]. 

The NR2 subtype NMDA receptor is ubiquitously distributed throughout the brain, with an unusually high density in the hippocampus [185]. The hippocampus is a brain structure that is linked to the autonomous nervous system, with critical importance for memory formation and learning, and it is also affected in mood disorders and in SS [93,94,186,187]. Likewise, the N-methyl aspartate receptor 2B (NR2B) subunit of the NMDA receptor is widespread in the dorsal root ganglion and it may mediate peripheral sensitization and visceral pain [188]. Those receptors are critically involved in the initiation and maintenance of neuronal hyperexcitability after noxious events and by C-fiber neuron stimulation, which consist of unmyelinated sensorial neurons [189]. 

In a rodent model of peripheral nerve injury subjected to tibial and peroneal nerve sectioning, leaving the sural branch nerve intact, with the aim to investigate the KP, the following changes were documented. After seven days, IDO1 was activated and kynurenine rose in the bloodstream, accompanied by depressive behavior (measured by an extended time of immobility in the forced swim test) and allodynia (tested by paw withdrawal in response to mechanical stimulation with von Frey hair). These findings were followed by an increase in the levels of KMO, quinolinic acid, and a reduction of kynureneic acid in the contralateral hippocampus [53]. These observations identify hippocampal neurons as the CNS site that is responsible for the perpetuation of pain and depressive symptoms. Injection of interleukin-1β (IL-1β) receptor antagonist in the CNS ventricular space reduced the depressive behavior and KMO mRNA levels, but did not change the allodynia, revealing the role of IL-1β in depression but not in the pain mechanism [53]. 

Depression has also been associated with decreased levels of tryptophan in patients with cancer receiving IL-2 and IFN-α therapy, suggesting that these cytokines impact the levels of serotonin [169]. 

Studies investigating the triggers of inflammation of the macaque CNS with poliovirus inoculation have revealed that quinolinic acid, kynurenine, and other metabolites of the KP accumulate in the spinal cord and CSF (but not as much in the bloodstream) at levels at those sites that are associated with the neurological manifestations [12,148]. Moreover, the in vitro conversion of L-tryptophan to kynurenine by fetal neuronal cells is dependent on IFN-γ stimulation in the presence of macrophages in culture [12]. Moreover, another study has shown that chronic pain in rats that are exposed to social stress or paw arthritis increases the levels of IDO and kynurenine and decreases the levels of serotonin in the hippocampus [190]. This situation is similar to that observed in human plasma levels of IDO, kynurenine/tryptophan and serotonin/tryptophan levels in patients with back pain and depression, in whom the first two rise and the third one decreases, as revealed in the same report [190]. Moreover, IDO1 knockout mice present reduced nociceptive and depressive behavior as compared with the wild type, and this behavior is not attenuated in the wild type that received an intraperitoneal injection of the NSAID acetaminophen, suggesting that this behavior is not dependent on the inflammatory mechanisms alone. Furthermore, the authors found that IL-6 is overexpressed in rats with arthritis and the Jak2/Stat3 signaling pathway is activated in the blood and hippocampus of rats with depressive and nocioceptive behaviors. Injection of IL-6 anti-serum attenuated the allodynia and hyperestesia in those animal models. In cultured Neuro2a cells (a mouse neuroblastoma cell line), incubation with IL-6 induced an increase in IDO1 mRNA and protein [190]. 

Therefore, persistent pain with allodynia and hyperalgesia are central components (spinal and supraspinal cord) supporting the involvement of glutamatergic neurotransmission associated with KP signaling in clinical manifestations and the role of the hippocampus as a critical organ in this process and the KP in the related physiopathology [9,30,41,190,191]. 

### Role of the Hippocampus in the KP in Neurological Manifestations 

Despite the agreement among clinical studies on changes in the KP and SS, it is admissible that the lack of an association between the symptoms of depression and fatigue in SS and the changes in the KP is due to difficulties in accessing and monitoring the changes in the CNS, more specifically in the hippocampus [18,56,192,193]. 

Not only SS patients, but also individuals exposed to chronic stress, present changes in the hippocampal structure and NMDA signaling [40,194]. Animal model studies have revealed that the hippocampus initially adapts to early, high, and frequent stress, but the persistence of aggression increases the levels of glutamate and disrupts hormonal and neurotransmitter control, leading to NMDA-driven neuronal death and hippocampal atrophy [195,196]. 

Moreover, hypothalamus-pituitary-adrenal activity has a modulatory effect on those events and the female sex hormone estrogen increases the synaptic connections and expression of nerve growth factor (NGF) in the hippocampus, supporting the conclusion that hormones influence and increase symptoms and pain sensitivity in females with SS [19,125,195,197].

Cytokines, such as IFN-γ, IL-1, and TNF-α, lead to increases in the expression of kynurenine and its metabolites, which are generated from tryptophan by the IDO enzyme, deviating this amino acid from the production of serotonin in the CNS throughout the KP (Figure 4) [13]. The resulting imbalance between the kynurenine metabolites and serotonin production in the hippocampus induces depression, slow reactions, and other cognitive disorders [198]. The target cells are microglia, astrocytes, and other inflammatory cells that are present in the hippocampus and other CNS areas [33]. Once impacted by those cytokines, the cells reduce glutamate reuptake, increase glutamatergic signaling, reduce the capacity to produce serotonin, trigger nociceptive and depressive behavior, and induce cell death (mostly the astrocytes), prolonging the inflammatory effect [13,33,41,53,199]. 

It is interesting to note that serotonin functions as a modulator of glutamate actions. PNS sensory transmission has silent glutamate synapses that are activated by serotonin. Once those silent synapses are activated by serotonin, they amplify the peripheral nociceptive glutamate signaling through the NMDA or α-amino-3-hydroxy-5-methyl-4-isoxazolepropionic acid receptor (AMPA) receptors from the spinal dorsal horn to the CNS [200,201]. In the CNS, glutamate/serotonin co-neurotransmission has been extensively studied, including in the hippocampus. The five subtypes of serotonergic receptors are expressed in different combinations among several cells, antagonizing the glutamatergic NMDA receptors at different levels, from preventing cell glutamate release to competing for the same intracellular signaling pathways in the hippocampus but not in other brain tissues [202,203,204,205,206,207]. 

Additionally, to demonstrate the differences in KP activity between the CNS and other parts of the body, systemic treatment with dexamethasone to reduce the inflammation that is induced by lipopolysaccharide (LPS) intraperitoneal injection was shown to promote a decrease in IDO enzymes in peripheral tissues (lung, spleen and liver) but an increase in brain microglial cells and astrocytes [199,208]. Moreover, the use of systemic subcutaneous slow-release corticosteroid pellets in rats increased the levels of NR2 NMDA glutamatergic receptors mRNA in the hippocampus [209]. These observations suggest that the use of corticosteroid treatment to reduce chronic inflammation may contribute to the nociceptive and depressive behavior over the long-term. 

Additionally, in human immunodeficiency virus (HIV)-infected patients, quinolinic acid (a metabolite of kynurenine that mimics the glutamate in NMDA receptors) content is several-fold higher in the brain than in the cerebrospinal fluid or blood [210]. 

Taken together, these findings delineate the hippocampus as primarily responsible for nociception and mood control, as well as a site where inflammation, driven by cytokines (mostly IFN-γ), induces a rise in KP activity and its metabolites (e.g., quinolinic acid and glutamate) with neuroactive actions to induce pain and depressive behavior. Moreover, during chronic inflammation such as in SS, the levels of serotonin in the CNS are diminished by tryptophan consumption throughout the KP. Reversion of the inflammation with corticosteroids in the CNS has been unsuccessful as the inflammation is present in other target organs and leads to the death of microglia, astrocytes, and neurons, mostly in the hippocampus and dorsal ganglion root [199,208,211]. These observations support the possible mechanisms of SS neurological manifestations, in which symptoms of pain and depression (i.e., allodynia, hyperalgesia, and fatigue), manifestations of reduced tear and saliva secretion, and elevated expression of blood markers of inflammation present a dissociation or discrepancy in the affected patients. Additionally, activation of the KP provides metabolites that mimic neurotransmitters and could be the cause of this dissociation and corroborate our hypothesis, as stated at the beginning of this review (Figure 4).

## 5. KP and Neuropathy in SS

Decades ago, reports have revealed that the ingestion of L-5-hydroxytryptophan induces signs and symptoms that are similar to scleroderma, with high plasma levels of kynurenine [212]. Moreover, excessive doses of tryptophan (greater than 1.2 g/day) trigger eosinophilia, severe muscular weakness and pain, and oral ulcers, with a rise in the hepatic enzymes aspartate and alanine aminotransferase (ASA and ALA), in addition to an inflammatory infiltrate in various organs [175]. Those events were associated with the upper plasma levels of kynurenine, as observed in cases with scleroderma and SS [153,212]. 

Salivary gland (SG) ductal ligation in rats induces tissue damage and atrophy, increases in systemic levels of kynurenine (as measured in the hair), and is associated not only with salivary hypofunction, but also body weight loss, over the six months of the experimental period. These findings indicate that higher levels of plasma kynurenine can reflect peripheral organ damage but also that SG damage is sufficient to impact whole-body metabolism, as demonstrated by the lower body weight as compared with the controls [21]. 

As mentioned above, the availability of the major substrate of the KP, tryptophan, is dependent on dietary intake, intake, but it is also influenced by environmental conditions and metabolism based on an individual’s genetic background [51,173,174,175,213,214,215]. For example, when female C57 ovariectomized mice ingest bisphenol A (BPA), an environmental contaminant with endocrine disruption capacity, it causes bowel inflammation and reduced levels of tryptophan and serotonin, indicating that environmental contaminants and the intestinal microbiota affect the KP in chronic inflammatory diseases [174].

Flow cytometric analysis of the peripheral blood has revealed a higher expression of IDO in the dendritic cells of pSS patients and in each of the subgroups, classified either by the presence of clinical or serological activity, as compared to the dendritic cells of healthy controls [216]. 

Measurements of IDO in antigen-presenting cells (APCs) and in T cells have demonstrated higher levels in the peripheral blood cells of pSS patients than controls matched by age and sex, also while using specific antibodies and flow cytometry, despite the heterogeneity of the groups and the high internal variability of the results [217]. Therefore, the T cell-mediated autoimmune activity present in autoimmune diseases, including SLE and SS, has been associated with elevated activity of the KP; however, the effects on autoantigen stimulation and IFN-γ activity remain unknown [216,217]. 

Different profiles regarding IFN-γ activity have been identified in the pSS population, and 55 genes and 19 metabolic pathways have been distinctly identified in a subset of pSS patients with fatigue [218,219]. Elevated IDO activity has been detected in IFN-γ-positive pSS patients, with higher levels of IDO mRNA and IFN-γ mRNA in circulating monocytes, and those observations were associated with the upregulation of apoptotic and neurotoxic downstream steps in the KP [18]. The levels of serum tryptophan are higher in healthy than in primary SS women (pSS). However, the levels of kynurenine and the kynurenine/tryptophan ratio are more elevated in pSS than in healthy women and patients with non-SS *sicca*. The same observations were found in pSS men, confirming the elevated activity of the IDO enzyme in the KP [153,220]. Moreover, the higher levels of kynurenine were associated with higher levels of inflammatory markers in the serum, such as the erythrocyte sedimentation rate, C-reactive protein, creatinine, Immunoglobulin A (IgA), β-2 microglobulin, and anti-nuclear antibody positivity [153,220]. Higher levels of kynurenine have also been associated with a lower proportion of individuals on corticosteroids but not with the frequency of neurological manifestations in the pSS group [153,220]. In another recent study, polyneuropathy showed more frequent positivity for the autoantibodies anti-Ro (SSa) and anti-La (SSb) in pSS [16]. Taken together, these studies suggest that the higher activity of the KP is related to clinical and laboratory signs of systemic inflammation, which may conflict with our hypothesis, but also indicates that those studies documented a midway point between the pain/neuropathic and the inflammatory poles of the disease [153,220,221]. Although the association of neurological or laboratory findings and kynurenine metabolites is evident in those studies, the cause/effect relationship between the metabolites of the KP and these manifestations remains unclear. 

In another study, an association of fibromyalgia and of other psychological symptoms, such as anxiety, depression, insomnia, psychoticism, and neuroticism, with fatigue being observed in a large series of pSS patients comprising 106 cases, among which 32 were fatigued and 74 non-fatigued, as identified by the Functional Assessment of Chronic Illness Therapy-Fatigue (FACIT-F) scale, with a cut-off of 30 on a scale ranging from 0 to 52. However, the levels of IDO mRNA in peripheral blood leukocytes did not differ between pSS patients with and without fatigue [192]. In addition, no other clinical or laboratory association was identified, excluding the number of individuals using hydrochloroquine, making up 50% of the pSS fatigue group and 28% of the pSS nonfatigue group. In contrast, the higher expression of IDO-1 mRNA levels has been associated with plasma levels of IFN-γ [192]. 

Fatigue has been associated with high KP activity levels in SLE patients, but only in those with clinical activity of the disease, as measured by the Systemic Lupus Erythematous Disease Activity Index (SLEDAI) with a score above five, which may confirm the possibility that the KP is overexpressed in the presence of elevated inflammatory activity [50]. Moreover, the levels of serum tryptophan are lower in SLE than in controls, systemic sclerosis (SSc), and pSS patients, which may reflect the higher activity and broad manifestations of the disease as compared to the other two conditions (i.e., pSS and SSc), which are more tissue-specific than SLE [193]. Interestingly, a modular IFN signature, including INF-α, INF-β, and INF-γ, has been identified in the majority of cases in a series of consecutive SLE patients [222]. Moreover, a metabolomic analysis comparing serum samples of SLE, pSS, and SSc patients and healthy volunteers allowed a specificity classification of 67% of the SLE group when compared with the other three groups, in which the most discriminatory metabolite was tryptophan, with lower levels in SLE than in the other groups [193]. The authors interpreted this reduction of tryptophan as a response to KP activation [193]. Experimental studies mimic the clinical findings of cognitive impairment, but not of depression, associated with the activation of microglia and astrocyte in the hippocampus in SLE mouse models induced by the injection of anti-ribosomal antibodies in the CNS as compared to controls [223]. Unfortunately, the KP was not investigated in that study. 

The evidence collected thus far from the medical literature does not confirm that the KP is the one and sole pathway responsible for the neurological manifestations of SS. In fact, two studies have presented opposing evidence, with higher KP metabolites being associated with higher levels of inflammatory markers [153,220]. This result may be due to (a) the heterogeneity of the SS cases recruited for clinical studies in terms of demography and disease duration; (b) the absence of a control group with healthy individuals in some studies; (c) alterations in the levels of metabolites with neuroactive properties in CNS tissues by KP activity-induced neurogenic changes, especially the hippocampus and dorsal ganglion root, but not in the blood, where they are measured in the actual studies; and, (d) observations collected in the middle of the inflammatory process and thus not representative of the anti-inflammatory effect of KP stated in the hypothesis. Such pitfalls must be taken into consideration in future studies addressing the present hypothesis of the association between SS neurological features and the KP.

## 6. Therapy to Modulate the KP

The overexpression of IFN-γ, induction of pro-inflammatory genes, such as TNF-α, interleukins, and B-cell activating factor (BAFF), the promotion of B cell activation and rise in autoantibodies in the blood are involved in the physiopathology of SS [99,156,224,225,226,227,228]. Therefore, therapeutic strategies to treat SS include immune modulators and biological therapy to inhibit B and T cells activity and proliferation [229,230,231,232]. The limitations of such strategies open opportunities for new procedures and complementary therapies. Among several possibilities, the potential modulation of KP has been explored in SS, as in other autoimmune diseases, including rheumatoid arthritis, SLE, and systemic sclerosis [15,233]. Experimental studies showing the effects of inflammatory challenges on the levels of KP metabolites and subsequent functional responses of the therapeutic modulation of this pathway in pain and neurological parameters are not always synergic and positive, which may be a consequence of observations that were collected at different time points and rebound effects [105,142,234,235,236,237,238,239,240,241,242,243].

Considering the broad spectrum of substrates, target tissues, and alternative effects of KP, therapeutic strategies to interfere with different steps have resulted in one or more of the following outcomes: attenuation of inflammation, reduction of chronic pain or improvement of fatigue, and depressive feelings in SS [18,217,233]. 

The traditional nonsteroidal anti-inflammatory drugs (NSAID), acetylsalicylic acid (ASA or aspirin), and sodium diclofenac have been investigated [244,245]. In rats, systemic intraperitoneal injection of tryptophan alone was able to increase kynurenine levels between 20 and 120 min. However, after combined injection of subcutaneous diclofenac with intraperitoneal tryptophan, the concentration of kynurenine increased, not just in the plasma and liver but also in the spinal cord and brain, with a remarkable increase in the kidney, after 60 and 120 min. Therefore, diclofenac disrupts the renal clearance of KP metabolites, which may amplify anti-inflammatory and excitatory stimuli on nociceptive NMDA receptors, independently of the analgesic effects of prostaglandins [245]. In an opposite manner, a study using human peripheral blood mononuclear cells (PBMCs) has revealed that aspirin at the dose of 5 mM, incubated for three days or 2 h, reduces tryptophan metabolism and kynurenine production in those PBMCs stimulated by concanavalin A and pokeweed mitogen (PKM), suggesting an inhibitory effect on IFN-γ based on the triggered mechanisms of action [244]. These observations indicate that NSAIDs have functions in addition to their known effects on the cyclooxygenase/prostaglandin pathway, and the impact on KP can be diverse, depending on the cell type and the specific NSAID [15]. It is also interesting to note that aspirin, a longer and broadly used analgesic drug in the NSAID group, has well-known positive effects on dry eye symptoms and LG dysfunction, which are critical elements in the manifestation of SS [246,247,248].

In an in vitro study using samples of T cells from 68 SS patients, coculture with mesenchymal stem umbilical cells revealed the suppression of proliferation and activation of these circulating follicular T helper cells, in association with the enhanced expression and enzymatic activity of IDO, as measured by reversal transcription polymerase chain reaction (RT-PCR) and high performance liquid chromatography (HPLC), respectively [249]. Another study has shown that human complementarity determining region 1 (hCDR1), a tolerogenic peptide that is complementary to the human anti-DNA monoclonal antibody, reduces the expression of inflammatory cytokines, down regulates the proliferation and activity of B cells, and increases the expression of anti-inflammatory cytokines in a rodent model of SLE [233]. In cultured mature leukocytes from 16 SS individuals, hCDR1 has been shown to reduce the expression of inflammatory cytokines, including IFN-γ, and increase the expression of anti-inflammatory cytokines, up regulating IDO gene expression. However, in the presence of 1mT, an IDO inhibitor, the effect of hCDR1 on the gene expression of the T cell regulator cytokine fork head box protein-3 (FOXP3) is reduced, suggesting that the immunomodulatory effect of hCDR1 is partially associated with its effect on IDO [233]. 

Suppression of the KP by inhibiting the KMO can reduce the pain that is triggered by LPS injection in the dorsal ganglion root in rodents. Systemic administration of the antibiotic minocycline or local administration of inhibitors of KMO [105] reduces the local levels of pro-inflammatory cytokines in the dorsal ganglion root and spinal cord and decreases pain and the protein expression of the following inflammatory mediators: ionized calcium-binding adapter molecule 1 (IBA-1), IL-6, IL-1β, and Nitric oxide synthase 2 (NOS2) [105].

When considering the strategy to overload KP to modulate the pain sensation, rats were subject to systemic administration of L-4-chlorokynurenine [250]. The experiments revealed that L-4-chlorokynurenine, a NMDA/glutamate receptor antagonist, administered by intraperitoneal injection reached the CNS and attenuated the hyperalgesia in four models of pain and behavioral response (general behavioral, formalin plantar injection, Carrageenan model, and Chung neuropathy) as compared with the controls, Dizocilpine (MK-801), and gabapentin [250].

Twelve of 20 GVHD individuals, who were non-responders to corticosteroids, presented clinical improvement in skin inflammation with human chorionic gonadotropin (hCG) treatment [251]. They also showed a significant increase in IDO mRNA expression in PBMCs and in IL-10 expression in the blood serum [251]. The underlying mechanism of action was thought to occur via stimulation of IDO-mediated immunotolerance, similar to the mother/fetus coexistence [144]. 

Although useful in its conception and to revert immune-mediated diseases, KP may be deleterious in neoplastic diseases, where it can allow tumor growth by inducing IFN secretion, and, after activating IDO, suppress the immune response toward the tumor [252]. According to this concept, the first human clinical trial aiming to block KP as an anti-cancer therapy was recently published [253]. The study investigated whether orally administered epacadostat, an IDO inhibitor, would be well tolerated and capable of slowing the growth of tumors in 52 refractory cancer patients by removing the immune tolerance to those tumors. The drug, at doses of 200 mg/day, reduced kynurenine plasma levels, indicating a reduction of tryptophan degradation. The mean treatment duration was 52 (from 7 to 284) days, and the daily doses ranged from 43 to 1400 mg. The side effects included fatigue, nausea, and pain, among others. However, no plasma changes in C-reactive protein or in the levels of the tested interleukins were observed [253]. The data confirmed the expected manifestations of KP inhibition and a safe strategy overall. Further conclusions are limited due to the small number and heterogeneity of the clinical cases.

Therefore, the present data on interventions in the KP reveal dual direction activities, in which inhibition at specific steps, such as KMO activity and quinolinic acid formation, has beneficial effects on neuropathic pain and neurodegenerative disorders, whereas enhancing the activity of IDO ultimately leads to an inhibition of pro-inflammatory cytokines and the reduction of inflammatory processes. How and at which steps those events can reach a conciliatory mechanism to diminish chronic pain and neurological symptoms in SS patients, but also prevent chronic inflammatory reactions will be the subject of further investigations.

## 7. Future Perspectives

IFN-γ triggers the deviation of the tryptophan to the KP pathway, possibly contributing to depression and pain through its action on particular organs, such as the hippocampus. This signaling pathway modulates the inflammation and chronic damage of the PNS and CNS, with potential repercussions on the SG, joints, and LFU. The suppression of APCs and production of anti-inflammatory cytokines in exocrine glands and other target tissues are the potential benefits of KP actions. Therefore, the elevated expression of IDO and kynurenine metabolites in SS suggests, but to date does not clearly indicate, a reactive process to modulate the mechanism of inflammation induced by other pathways. Improved strategies to access the CNS and PNS organs by imaging analysis and to monitor the local activity of the KP in the involved organs in SS, glandular or not, would facilitate insights regarding the physiopathology of this signaling pathway in SS, its interference in exocrine secretion impairment and strategies for improvement. More effective treatments and an enhanced quality of life for SS patients will be potential benefits from this knowledge.

## 8. Conclusions

Glandular and EGM of SS are not exclusively inflammatory but also involve a neuroimmunendocrine network, in which the KP plays a role. The involvement of the KP is difficult to track and confirm because of the delicate methods for tracing, the timing of the response, and the location of the metabolites of this pathway in the CNS. A better understanding of the relationship between the physiopathology of SS and the KP in the CNS and target tissues may help to clarify the discrepancies among the signs and symptoms and the neurological manifestations. This knowledge could improve therapy for SS. 

## Figures and Tables

**Figure 1 ijms-19-03953-f001:**
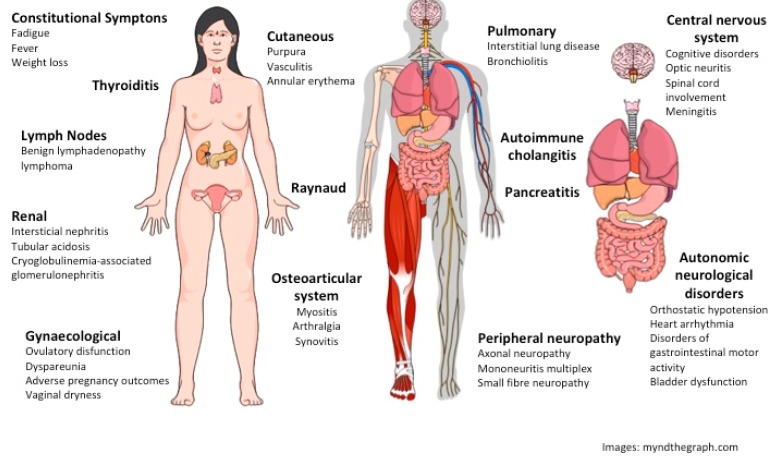
Extraglandular manifestations of Sjogren’s Syndrome. The pleiotropic features of systemic multiorgan involvement in SS are not well-understood. Some of the extraglandular manifestations may arise from immune-complex deposition in the context of cryoglobulinemia. Other symptoms and signs are related to lymphoproliferation, dendritic cell activation, and cytokine maintenance of the inflammatory process. Interferon-γ-inducible-Kynurenine Pathway could play a role in the neural manifestations, fatigue and chronic pain [16]. Figures were obtained from the free version of myndthegraph.com.

**Figure 2 ijms-19-03953-f002:**
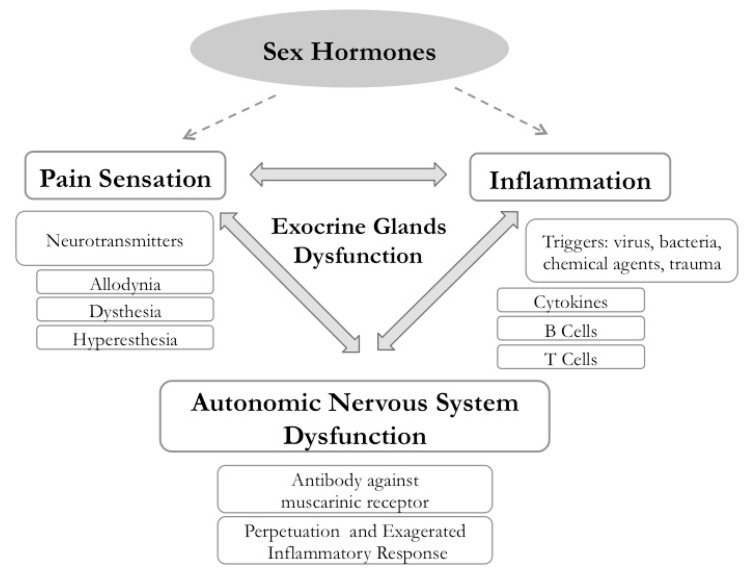
Schematic model showing the interrelationships among autonomic nervous system dysfunction, pathological pain and chronic inflammation in Sjögren’s syndrome (SS) [19,29]. The dotted arrows indicate the effects of hormones with all their cyclic, indirect and also circadian influences. The gray two-way arrows indicate the interdependence of the three phenomena highlighted in the boxes: pain sensation, inflammation and autonomic nervous system dysfunction.

**Figure 3 ijms-19-03953-f003:**
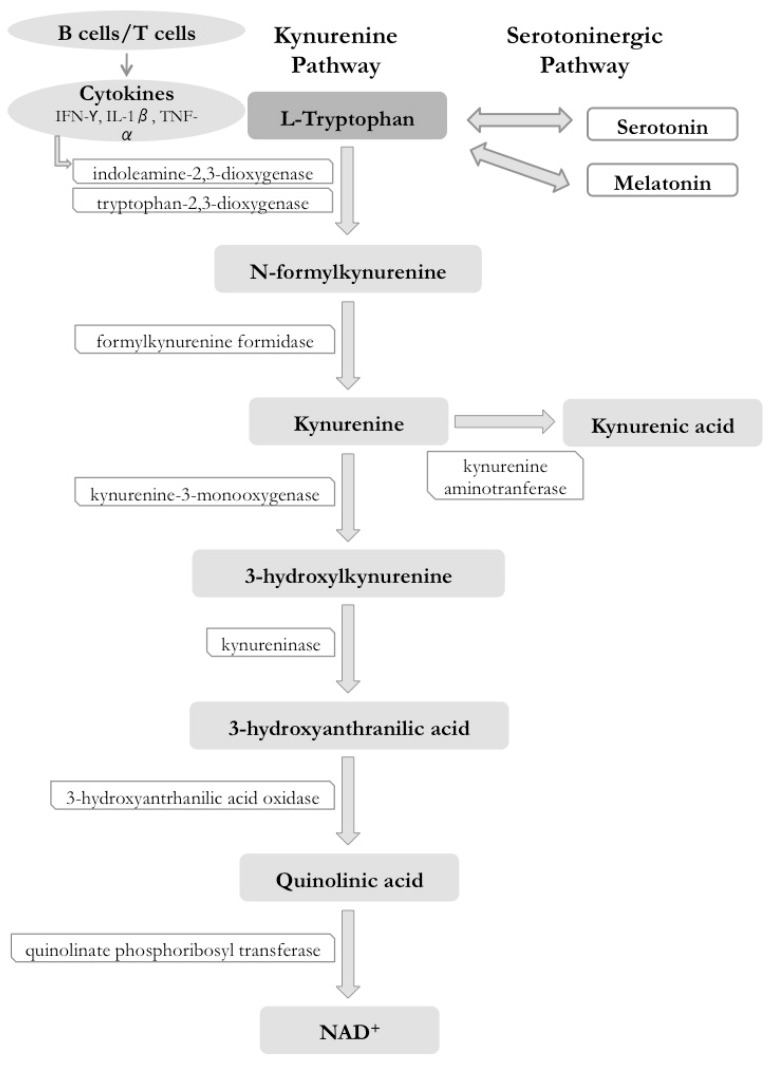
Kynurenine pathway (KP) and metabolites. Several of the metabolites have neuromodulatory effects [14,140,141,142]. The fine gray arrows indicate chemotactic and enzymatic effects, the thick gray arrows indicate the subsequent metabolite in the cascade, whereas the two-way gray arrows indicate the potential conversion of one molecule in the other, depending o demand and the subtract availability. IL-1β is Interleukin -1beta and TNF-α is Tumor Necrosis Factor-alpha.

**Figure 4 ijms-19-03953-f004:**
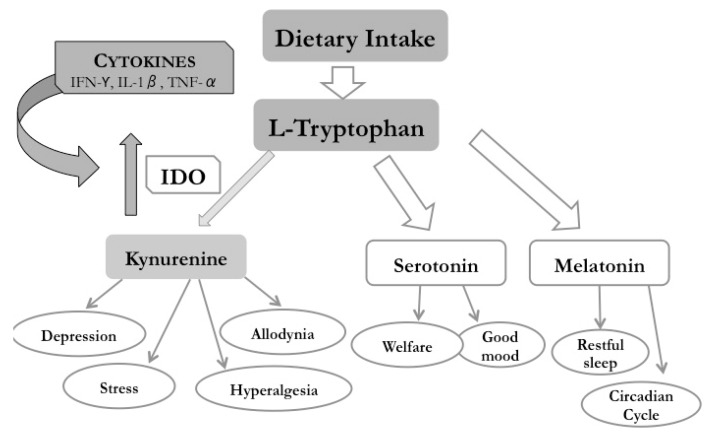
Sjögren’s syndrome physiopathology, considering the elevated cytokine expression activation of the kynurenine pathway (KP) and its implications in the pain and mood symptoms dissociated with the signs of SS disease. The light thick gray arrows indicate the pathway to obtain L-tryptophan to produce serotonin and melatonin and the dark gray arrow indicate the catabolic pathway of the L-tryptophan and its relationship with inflammatory cytokines. The thin gray arrows indicate the clinical manifestation of different L-tryptopahn metabolite.

**Table 1 ijms-19-03953-t001:** Common locations of changes in the nervous system and in the exocrine glands caused by experimental traumatic modulation of the kynurenine metabolic pathway (KP) and Sjögren’s Syndrome.

Structure	Kynurenine Metabolic Pathway	Sjögren’s Syndrome
Dorsal ganglion root	Sciatic injury increases kynurenine monooxygenase (KMO) in the dorsal root ganglion and spinal cord of rats [105]	Dorsal root ganglion alterations in MRI, associated with increased intradermal nerve fiber density on skin biopsy [109]
Hippocampus	IDO and kynurenine-3-hydroxylase increase in the hippocampus after day 2 after CNS ischemia [110]	Hippocampal atrophy in SS patients [40]
Exocrine Glands and LFU	Increase in kynurenine in salivary gland after ductal ligation, LG atrophy due to tryptophan deprivation [20,21]	Changes in LG and SG in the MRI, nerve changes in the cornea of SS patients [107,111]

MRI: magnetic resonance image, SS: Sjögren’s syndrome, LG: lacrimal gland, SG: salivary gland, CNS: central nervous system, IDO: indoleamine 2,3-dioxygenase.

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
