# Peer review of "Neurological and Inflammatory Manifestations in Sjögren’s Syndrome: The Role of the Kynurenine Metabolic Pathway"

_ijms, 2018, doi:10.3390/ijms19123953_

Reviewer 1 Report

This is a comprehensive review of the immune- and sensorineural modulatory effects of the Tryptophan/Kynurenine pathway (TKP). The authors highlight the complex interaction between the immune, endocrine and nervous systems in autoimmune conditions, like Sjogren’s syndrome (SS). They offer a hypothesis that TKP is involved in the mechanism of SS neuropathy, since compensatory crosstalk pathways can drive the dissociation between signs and symptoms among the immune, the endocrine and the neural systems. They provide a detailed description of the TKP and interactions of tryptophan metabolites with neuropeptides/receptors in the central and peripheral nervous system. Like many aspects of SS, the findings are often contradictory and difficult to interpret. The authors have highlighted this throughout the review. 
1. The review makes a good case that the TKP is relevant, but it doesn’t really address the dissociation between signs and symptoms. Perhaps they can clarify what this means with regard to patients with many symptoms and lack of signs, or patients with many clinical manifestations or high inflammatory markers and minimal signs. A summary figure addressing this would be helpful. 

2. Additionally, the manuscript would benefit from thorough review and editing by an English fluent reader because there are numerous grammatical errors and the meaning of some sentences are not clear. 

Specific points

3. Box 1. The term “aggression” is used in a number of sentences throughout the manuscript. This is not a conventional term and the meaning is not clear so it should be replaced. 

4. Section 2, lines 71-74. This sentence is not clear and should be rewritten

5. Section 2.1, line 87 “Prevalence of” should be added before peripheral neuropathy in sentence 2

6. Figure 2. Arrows indicating the neuromodulatory effects of relevant tryptophan metabolites should be added

7. Section 3, lines 200-201. Expand on the relationship between interferon alpha and gamma (synergistic vs antagonistic)

8. Section 5. Lines 401-403. It is not clear that this sentence disagrees with the hypothesis, please revise for clarity (i.e. how does it disagree?)

9. Table 3. The results of the preclinical studies shown in Table 3 report that inhibition of IDO worsens clinical signs/inflammation. Is it unreasonable to assume that if IDO promotes disease resolution there would be a subsequent decrease in IDO and neurostimulatory effects of TKP? 

 Author Response
1. It is not proven that the dissociation between signs and symptoms are caused by an unbalance in the kynurenine versus serotonin levels. Our hypothesis is that kynurenine metabolites are working as neurotransmitters enhancing the symptoms of pain and depression. To clarify this point we rephrased the following sentence and the legend of Figure 3, as follow (page 25):

These observations support the possible mechanisms of SS neurological manifestations, in which symptoms of pain and depression (i.e.; allodynia, hyperalgesia and fatigue), manifestations of reduced tear and saliva secretion and elevated expression of blood markers of inflammation present a dissociation or discrepancy in the affected patients. Additionally, activation of the KP provides metabolites, which mimic neurotransmitters and could be the cause of this dissociation and corroborate our hypothesis, as stated at the beginning of this review (Figure 4).

Figure 4. Sjögren’s syndrome physiopathology, considering the elevated cytokine expression activation of the kynurenine pathway (KP) and its implications in the pain and mood symptoms dissociated with the signs of SS disease.

2. As mentioned above, the manuscript was submitted to professional English edition and the certificate is attached to this submission.

3. The term was replaced by “inflammation”.

4. The sentence was rewritten as follow and references were added (page 7):

The dissociation between signs and symptoms in dry eye and dry mouth disease, including SS, delays and makes the diagnosis more difficult [5, 42-48]. The lower intensity of signs associated with depression and diffusely spread painful symptoms in the body suggest a mechanism involving integrative systems such as the autonomic nervous system, immune and endocrine systems (Figure 1) [42, 45, 49].

5. The expression was added and highlighted.

6. Due to the numerous and still growing metabolites with descriptions of neuro effects we opted for adding the info and references to the Figure 2 legend and keeping the figure simpler. As follow:

Figure 3. Kynurenine pathway (KP) and metabolites. Several of the metabolites have neuromodulatory effects [14, 140-142].

7. To clarify the antagonic effect of tyoe I and II INF in SS, we rephrased the sentence as follow (page 19):

IFN-α and β are associated with inflammatory activity (including higher levels of autoantibodies and inflammatory cell activity), and IFN-γ is associated with TKP activation and the attenuation of inflammatory activity [91, 159-161]. SS patients have higher levels of IFN-γ and INF-α and β (type I INF) mRNA in the peripheral blood and labial salivary glands in comparison to healthy individuals, and SS patients with lymphoma present lower levels of IFN-α and higher levels of INF-γ mRNA in labial salivary gland samples compared with other SS and control individuals, supporting INF-α/γ as a predictor of lymphoma in SS [156]. Therefore, type I IFN seems to have an antagonistic effect to type II IFN but an unclear, though potentially useful, combinatorial effect in SS associated with lymphoma.

8. The paragraph was rephrased as follow to improve clarity (page 32):

The evidence collected thus far from the medical literature does not confirm that the KP is the one and sole pathway responsible for the neurological manifestations of SS. In fact, two studies have presented opposing evidence, with higher KP metabolites associated with higher levels of inflammatory markers [153, 220]. This result may be due to a) the heterogeneity of the SS cases recruited for clinical studies in terms of demography and disease duration; b) the absence of a control group with healthy individuals in some studies; c) alterations in the levels of metabolites with neuroactive properties in CNS tissues by KP activity-induced neurogenic changes, especially the hippocampus and dorsal ganglion root, but not in the blood, where they are measured in the actual studies; d) observations collected in the middle of the inflammatory process and thus not representative of the anti-inflammatory effect of KP stated in the hypothesis. Such pitfalls must be taken into consideration in future studies addressing the present hypothesis of the association between SS neurological features and the KP.

9. The preclinical studies show that blocking some steps benefit neurologic manifestations, including pain. The time-frame of events reporting lowering the TKP metabolites and the subsequent effects, needs further investigation. Some observations may represent the return to the mean” and/or rebound effects. To clarify this point, the following phrase was added to text (Session 6, page 33):

Experimental studies showing the effects of inflammatory challenges on the levels of KP metabolites and subsequent functional responses of the therapeutic modulation of this pathway in pain and neurological parameters are summarized (Table 3) The results are not always synergic and positive, which may be a consequence of observations collected at different time points and rebound effects (Table 3) [105, 142, 234].

Reviewer 2 Report

Overall, this is an interesting review on an intriguing topic. The authors have reviewed most of the existing literature in mouse models and human disease relating to the tryptophan/kynurenine pathway, and made allusions to its potential role in Sjogren's syndrome, including potential targets for novel therapies. However, there are major concerns with the manuscript as it stands:

1. The English editing leaves much to be desired and often gets in the way of reading and understanding the points the authors wish to make. It would need rigorous revising for grammar and style.

2. Hypothesis needs clarification - consider stating some rationale leading up to it, then stating the hypothesis. Is it that they want to investigate the role of interferon-gamma inducible TKP activity in the pathogenesis of the chronic pain and extraglandular manifestation disease activity in Sjogren syndrome?

3. If the answer to the question in 2 is 'yes', the rest of the paper needs re-organization and focus to do just that. As it stands, the entire review needs more focus - particularly in how ALL of the models reviewed relate to their hypothesis on Sjogren syndrome. If any of the models do not particularly contribute to some theory about Sjogren syndrome, I would consider removing those data from the paper. At times, the discussion about various models seemed to weigh down the paper.

4. Along the same lines, rigorous editing for content would be helpful - this review is repetitive in parts and some sections might be better combined.

5. Tables need revising for simplification ( table 1) and clarification (table 2). I am not clear on what this was meant to show or why they include 'TKP modulated' - if these are from several research studies, say so, and give some background and a more descriptive legend.

6. Consider adding a section on the perspective of novel -omic methodology. This would give a fresh take on the data presented here and perhaps lend more credence to an association with Sjogren syndrome. For example, include data on what is known to date in metabolomic studies in autoimmunity - consider starting with what is known in other diseases - there was a paper in PloS One in 2016 comparing the metabolomes of SLE patients with those of Sjogren syndrome and systemic sclerosis patients.

I humbly apologize that I cannot, at this time, give a line by line edit on this paper due to the grammatical and stylistic errors.

Thank you for the opportunity to review this manuscript. Please do not hesitate to contact me with any questions.

 Author Response

1. The manuscript was submitted to a professional editorial assistance for English. The changes were revised. The certificate of this review is attached in this resubmission.

2. In order to clarify the hypothesis, we rewritten the Hypothesis paragraph as follow:

The pain, depression, and other neurological manifestations of SS can be associated with IFN-γ inducible KP activity. Therefore, our hypothesis considers three aspects: a) the KP in the mechanism of SS neuropathy; b) the possible roles of the KP in cross-talking pathways among the immune, endocrine and neural systems to produce the SS signs and symptoms and, c) the possible role of the KP in the previously reported dissociation between those signs and symptoms in SS [50-55].

In summary, this hypothesis implicates the KP in the spectrum of manifestations of SS, with two different poles of SS disease.

3. As observed by Reviewer 2, the hypothesis involves the pathogenesis of a complex disease with mosaic of manifestations and a signaling pathway involved in neurologic and immune diseases. We limited the comparisons to human studies in chronic pain, GvHD and SLE and

few studies using animal models that were able to indicate potential clues for SS.

Unfortunately, the paucity of studies focused in SS and TKP and the impossibility to find a strong confirmation of this hypothesis lead us to observe the related literature. Redundant and excessive information were removed.

4. The text was revised by the corresponding author, to obtain a more concise and a clear manuscript.

5. The table 1 was revised for simplification. The text and the table legend was rewritten to improve clarity of the Table 2, as follow:

It is interesting to note that structures that are altered in the CNS, PNS, exocrine glands and cornea (based on image exams) of SS patients are also sensitive to changes in the KP caused by trauma or ischemic damage (Table 2). These observations suggest that common inflammatory events are shared in autoimmune and non-immune inflammatory diseases, such as the synapses of the sensorial fibers, in the dorsal ganglion root, hippocampus, thalamus and LFU [2, 12, 13, 20, 29, 40, 53, 93, 105-108] (Table 2). Those locations must be investigated in future studies addressing the anatomic correlations of clinical and biochemical changes in SS.

Table 2. Common locations of changes in the nervous system and in the exocrine glands caused by experimental traumatic modulation of the kynurenine pathway (KP) and Sjögren’s Syndrome.

6. We agree that omics may help, but the discrepancies found here were pointed in our observations that were associated to the local organs affected in SS in contrast to the diffuse disease observed in SLE. To address this point we added the following phrase and the suggested reference:

Moreover, a metabolomic analysis comparing serum samples of SLE, pSS and SSc patients and healthy volunteers allowed a specificity classification of 67% of the SLE group compared with the other three groups, in which the most discriminatory metabolite was tryptophan, with lower levels in SLE than in the other groups [193].

Reviewer 3 Report

Excellent compilation of information.

1. A figure describing different extraglandular manifestations of SS will be helpful.

2. Dry mouth and dry eye are the primary symptoms of SS. A paragraph describing neurological control of saliva and tear production might be helpful.

 Author Response

1. Figure 1 was added and the subsequent figures were renumbered.

The legend is the following:

Figure 1. Extraglandular manifestations of Sjogren’s Syndrome. The pleiotropic features of systemic multiorgan involvement in SS are not wellunderstood. Some of the extraglandular manifestations may arise from immune-complex deposition in the context of cryoglobulinemia. Other symptoms and signs are related to lymphoproliferation, dendritic cell activation, and cytokine maintenance of the inflammatory process. Interferon-γ-inducible-TKP could play a role in the neural manifestations, fatigue and chronic pain [16]. Figures were obtained from the free version of myndthegraph.com.

2. The suggestion was included in the beginning of the session 2, page 6, as follow:

The constant basal and reflex wetting of the mouth and the ocular surface provided respectively by saliva and by tears are directly controlled by the autonomic nervous system [22-24]. The volume and content of fluids from several exocrine glands present in both locations (mouth and eye) are responsive to sensorial stimuli from the environment driven by sensitive nerves specialized in taste and vision but also general sense nerves related to touch, thermal and chemical changes [25-27]. Once detected by the brainstem, the feedback mechanisms are conducted in the parasympathetic and sympathetic systems, stimulating synapsis in α-adrenergic and muscarinic cholinergic receptors in the many salivary and lacrimal gland subtypes distributed in the mouth and in the ocular surface [25, 28]. This sensorial/autonomic feedback system that regulates the tear secretion in the ocular surface is called the lacrimal functional unit (LFU) (and a very similar system works in the mouth), whereas inflammation, trauma or other damage in a segment of this integrative system can disrupt persistent dysfunction of secretion with variable sensorial manifestations, depending on the integrity of the sensorial loop [28, 29].

Round  2

Reviewer 1 Report

The authors adequately addressed all the points in the review. 

Author Response

Thank you very much for your suggestions and comments. 

Reviewer 2 Report

This manuscript is much improved from previous. In particular, the grammar is no longer a hindrance to its comprehensibility. The added sections relating KP pathway back to autoimmunity and different manifestations of Sjogrens Syndrome make the article more cohesive.

The one major edit I would recommend is the removal of tables 1 and 3 as I find them cumbersome and tedious to the article. The authors do explain most of the studies within the text so the tables are a bit redundant.

Thank you for the opportunity to review this manuscript. I would be happy to review it again should the need arise.

Author Response

Dear Reviewer, 

We followed your suggestions and removed tables 1 and 3. We renumbered table 2 as 1 and edited the text to make it clear. 

Thank you very much for your suggestions and comments on our work.

Reviewer 3 Report

Good revision and very good effort

Author Response

Thank you very much.